# Promoting Subjective Well-Being among Rural and Urban Residents in Indonesia: Does Social Capital Matter?

Tri Wahyu Nugroho, Nuhfil Hanani, Hery Toiba * and Sujarwo Sujarwo

Agriculture Socio-Economic Department, Faculty of Agriculture, Brawijaya University, Malang 65145, Indonesia; tw.nugruho@ub.ac.id (T.W.N.); nuhfil.fp@ub.ac.id (N.H.); sujarwo@ub.ac.id (S.S.)
* Correspondence: htoiba@ub.ac.id

**Abstract:** There has been growing research on the link between social capital and subjective well-being. However, to date, research investigating the impact of social capital on subjective well-being based on urban and rural typology is limited. Therefore, to fill this gap, this study aims to examine the effects of social capital on subjective well-being, based on urban and rural typology, using large-scale data from 29,341 Indonesian residents, comprising 17,155 urban residents and 12,186 rural residents. A two-stage predictor substitution (2SPS) approach is applied to address the endogeneity issue in estimating the impact of social capital. The empirical findings indicate that social capital significantly increases subjective well-being, i.e., happiness and life satisfaction. However, based on the urban–rural model, we found that the impact of social capital on subjective well-being is different. In the urban model, social capital increases happiness and life satisfaction significantly. However, the rural model indicates that social capital significantly increases happiness, not life satisfaction. These findings imply that subjective well-being impacts urban residents more than rural residents. The main reason is social capital in urban areas is well-developed (i.e., management and infrastructure for community association). Therefore, we suggest developing social capital in rural areas to expand its role in improving well-being.

**Keywords:** subjective well-being; community associations; social capital; rural; urban; 2SPS; happiness; life satisfaction



## 1. Introduction

Over the last few decades, governments and researchers have made considerable efforts to reduce global poverty. It is no wonder that the first sustainable development goals (SDGs) of the United Nations focus on eradicating poverty. However, the United Nations claimed that, in 2020, the global poverty rate increased to around 119–124 million, suggesting that the efforts have not been fruitful. Researchers have been studying subjective well-being to reduce poverty, since it is a more reasonable measure of the level of personal welfare [1]. However, the findings have not been conclusive [2]. Moreover, Easterlin [3] postulated "happiness economics", which sparked a debate among scholars. The study on the link between income and happiness shows that, on average, those with more income claim better subjective well-being. However, as people grow wealthier, the level of happiness will stop increasing at some point. This phenomenon is known as the "Easterlin paradox". Bartolini and Bonatti [4] also pointed out that economic growth could reduce individuals' well-being by deteriorating social relations. Such impact on social relations—or social capital in general—needs to be examined to understand subjective well-being. Therefore, economic research has been exploring this area, especially since Putnam [5] discovered a strong relationship between social capital and economic indicators in the United States of America.

According to Bourdieu [6], social capital is "the aggregate of actual or potential resources linked to a possession of a durable network of more or less institutionalized

relationships of mutual acquiescence or recognition." Similarly, Putnam [7] defines social capital as "features of social life—networks, norms, and trust—enabling participants to act together more effectively to pursue shared objectives". These definitions emphasize the role of one's social relationships in accessing essential resources. Its core premise is that group involvement and participation can benefit both the person and the community if they are well connected. The advantages can also be non-monetary, that will eventually lead to favorable outcomes. Among these are interpersonal trust, a sense of belonging, and reciprocity [8]. The roles of social capital in economic sectors and society, in general, have been well documented in the literature. We group it into categories: macro- and microeconomics perspectives.

From the macroeconomic perspective, research has highlighted the impact of social capital on the economic growth [9–12]. For example, a study conducted by Muringani, Fitjar and Rodríguez-Pose [12] investigated the impact on regional economic development in 21 European countries. Using bonding and bridging as dimensions, they reported that social capital could be associated with high economic growth. Likewise, Makiyan, Karbalaei and Bahrami [11] compared the economic growth between countries with low and high social capital. The finding revealed that economic growth in countries with high social capital tends to be higher than those with lower social capital. Hörisch and Obert [10] also claimed a positive association between economic crisis and social capital.

Some research focuses on the microeconomic perspective [13–15]. For instance, Maluccio, et al. [16] examined the impact of social capital on households' well-being in South Africa, indicating a positive impact. However, it is critical to note that the estimation results vary according to the years. The estimation from the 1993 data set showed an insignificant effect, but the 1998 data set showed otherwise. This implies that the role of social capital has increased over time. Meanwhile, Wahyuni [14], using large-scale data from 24,175 Indonesian residents, found that social capital improved household well-being through better income and expenditure. More recently, Kairiza, Kembo, Magadzire and Chigusiwa [13] estimated how social capital impacts food security in Zimbabwe, showing better security status among households with the highest social capital.

As for research on well-being, it is now seen as a multidimensional notion encompassing numerous areas of life. Traditional measures, such as income, poverty status, and food security, have been challenged for being excessively one-sided and incapable of accounting for non-monetary aspects of quality of life, such as happiness and life satisfaction. As a result, scholars in allied fields such as economics and social science began to demonstrate how social capital affects subjective well-being [17–20]. For instance, Hommerich and Tiefenbach [17] used social affiliation as an indicator of social capital to examine its relation to well-being and found a positive and significant relationship. A similar investigation in Japan by Matsushima and Matsunaga [18] supports the finding and indicates that volunteering and trust significantly impact well-being, whereas organizational membership does not. However, a study in China found the opposite and stated that organizational membership correlates with well-being [21]. A tentative explanation of the different findings is socioeconomic conditions, especially if the gap is wide between rural and urban areas. Not only do the socioeconomic conditions vary, but so do the education, infrastructure, and technology innovation. More often than not, rural areas are left behind in all aspects.

Although research has extensively investigated social capital's impact on subjective well-being [17–21], these studies are general [17,18,20] and offer only regional perspectives [19]. To date, no specific research has investigated the impact's magnitude on rural and urban areas. Therefore, we aim to fill this gap by investigating the impact of social capital on subjective well-being based on rural and urban typology. Generally, this study is motivated to support the United Nations' SDGs, notably "no one should be left behind", by reducing urban–rural disparity and promoting well-being in all layers of society. Likewise, this study contributes to the literature by in three innovations. First, we applied two-stage predictor substitutions (2SPS) to overcome the endogeneity issue when estimating the impact of social capital on subjective well-being. Residents voluntarily participate in a

community association as the social capital indicator, influenced by observable factors, such as socio-demographic condition, and unobservable factors, such as motivation. As a result, the social capital variable is potentially endogenous and should be addressed in the estimation to avoid bias estimation [22]. Second, we extend the body of literature by providing an urban–rural typology model, which has not been covered in past research [17–21]. Estimating the effect of social capital by separating the model of urban and rural will provide a deep understanding of how social capital affects rural and urban residents' subjective well-being. Third, we employed national representative data to capture the whole Indonesian population.

The remainder of the article is structured as follows: Section 2 outlines the literature review; Section 3 outlines the research methodology; Section 4 provides the empirical results; Section 5 presents the discussion; Section 6 presents the conclusion of the discussion and policy implications.

## 2. Literature Review

In recent years, the importance of non-monetary components has become more acknowledged as well-being, which is now recognized as a multidimensional concept encompassing multiple aspects of life [23]. Traditional well-being measures—such as gross domestic product, as a measurement on a regional level, and income on an individual or family level—have also been challenged as being unduly one-sided and incapable of accounting for non-monetary aspects of life quality, such as happiness and life satisfaction [24,25]. As a result, experts in adjacent fields such as economics and political science started to take an interest in subjective well-being [3,26–29], defined as personal cognition and affective evaluation of life quality [30,31]. A publication by Diener, et al. [32] defines subjective well-being as "a broad category of phenomena that includes people's emotional responses, domain satisfactions and global judgments of life satisfaction." Past research has widely used two indicators to measure subjective well-being: happiness and life satisfaction [22,24,33,34]. Happiness represents the emotional quality of daily experience in the short run, whereas life satisfaction indicates long-term thoughts and feelings about life [35,36]. An array of variables associated with subjective well-being have been identified in the literature, such as socio-demographic characteristics (such as age and education), personality, social and natural environment [26,32], income and consumption [30], spatial, social, and cultural mechanisms, political processes [37,38], and technological adoption [22]. One of the essential factors promoting subjective well-being is social capital [17–20].

In this exploration, social capital is measured by the membership to six essential community associations, namely community membership, community meetings, cooperatives, voluntary work, village/neighborhood improvement programs, youth groups, and religious activities. These associations can be found in many developing countries across the globe [39], and they can improve subjective well-being through increasing health status, education, literacy, sanitation, financial capital, and individual or community relationships [40]. More access to the resources from networking in those activities is associated with higher social capital. Similarly, Kawachi, et al. [41] explain how a community's social capital influences the quality of life. To begin with, social capital acts as a conduit for the dissemination of well-being-related knowledge and information. Second, social capital can be used to preserve norms and impose social control over potentially harmful behaviors. Third, social capital enables access to services and amenities because cohesive neighborhoods are more adept in organizing collective actions to promote the development of and access to livelihood resources. Fourth, social capital facilitates psychosocial processes, such as forming social support and mutual esteem.

The impact of an association membership (as social capital's measurement) on subjective well-being has been documented in the literature, but the findings are contradictory. For example, a study by Haller and Hadler [42], which covers many countries in the World Values Survey, and a study by Yip, Subramanian, Mitchell, Lee, Wang and Kawachi [21] shows an insignificant relationship between community association and subjective well-

being. In contrast, Pichler [43] revealed that the number of civil society organizations significantly promotes subjective well-being in European countries, with residents being active members in communities reporting better subjective well-being. Research by Huang and Humphreys [44] in the United States found that membership boosts happiness. Furthermore, transitional countries, such as Colombia, observed a favorable link between its community association and the citizens' happiness [45]. Fancourt and Steptoe [46] also investigate the association between community memberships and subjective well-being, using longitudinal data from 2548 adults in England. The results indicate a positive association between membership and subjective well-being.

While there is clear evidence for the beneficial impact of social capital on subjective well-being in general, the effect of urban–rural disparities in social capital is less evident. Only a few pieces of research have examined differences in the magnitude of social capital's impact on subjective well-being in rural and urban settings. The first investigation by Mair and Thivierge-Rikard [47] compares the effect of social capital on subjective well-being among rural and urban residents. They discovered that informal social capital, such as the relation with friends, relatives, and neighbors, is greater for rural residents than for urban residents. Different expectations about social interactions could be the cause of these different outcomes. Individuals in rural areas may expect more casual social relationships than those in urban areas. Secondly, Zawisza and Tobiasz-Adamczyk [48] found no clear evidence of social capital effects on subjective well-being in urban and rural settings in Poland. Because the results for different regions in the research location vary, there are two research gaps. First, the study only captures informal social capital, namely relation friends, relatives, and neighbors. Past studies did not include the different effects of social capital from formal institutions (i.e., community association). Meanwhile, the development of community association is also different in urban and rural areas. Secondly, past research only involves older adults. Therefore, investigating the different effects of social capital, measured by community association, is essential in extending the existing literature [47,48].

## 3. Materials and Methods

### 3.1. Research Data

This study employs open access cross-sectional data from Indonesia Family Life Survey (IFLS-5), accessible at http://www.rand.org/labor/FLS/IFLS.html (accessed on 4 December 2021) The IFLS-5 is a population-based household survey conducted in 2014–2015, with a multi-stage, stratified sampling design (321 enumeration areas—EAs; 20 and 30 randomly selected households from each urban and rural area, respectively, in 13 out of 27 Indonesian provinces), covering 83% of the Indonesian population [49]. The survey was in English and translated into Bahasa Indonesian by survey professionals before being retranslated into English by two independent, outsourced translators. The survey was also pre-tested on 393 household members. The IFLS was authorized by RAND and Gadjah Mada University's ethics review committees. Since the IFLS survey contains plenty of information, we only kept the variables required for this study, namely social capital, socio-demographic profiles, and subjective well-being. After cleaning the data, respondents with missing data were eliminated. Finally, we consider 29,341 residents as our respondents, specifically 17,155 urban residents and 12,186 rural residents.

To employ an evaluation impact estimation, we follow Ma and Abdulai [50], Zheng and Ma [22], and Rahman, et al. [51], who divided the study's variables into three categories. The first is the interest or treatment variable, i.e., social capital. Following Calcagnini and Perugini [8], Yip, Subramanian, Mitchell, Lee, Wang and Kawachi [21], and Maluccio, Haddad and May [16], we consider community membership as an indicator of social capital. The community membership covers community meetings, cooperatives, voluntary work, village/neighborhood improvement programs, youth groups, and religious activities. We summarize the residents' participation in these six groups, with the value ranging from one to six. The second variable is the outcome variable, i.e., subjective well-being. Following Zheng and Ma [22], Ngamaba and Soni [34], Sujarwoto [49], and Blekesaune [33], we used

two subjective well-being indicators, namely happiness and life satisfaction. In the IFLS data, the happiness variable was obtained of the following question: "Taken all things together, how would you say things are these days—would you say you are very happy, happy, unhappy, or very unhappy?"; the ordinal response ranged from 1 for very unhappy to 4 for very happy. Meanwhile, the life satisfaction question was: "Please think about your life as a whole–how satisfied are you with it?"; the responses were measured in a 5-scale range, starting with 1 for not satisfied and 5 for completely satisfied. The third category was the control variables, including age, marital status, education, family members, and access to technology, such as TV and the internet [16,52–54].

*3.2. Estimation Strategy*

To estimate the impact of social capital on subjective well-being, we assume the subjective well-being measure is a linear function of social capital ($S_i$) and other control variables ($X_i$). Therefore, we can specify the model as follows:

$$H_i^* = aS_i + \beta X_i + e_i, \text{ and } H_i = \begin{cases} 1 \text{ if } H_i^* \leq C_1 \\ 1 \text{ if } C_1 <; \ H_i^* \leq C_2 \\ n \text{ if } C_{n-1} \leq H_i^* \end{cases} \qquad (1)$$

where $H_i^*$ is the latent variable indication the respondent's—$i$—subjective well-being level, including happiness and life satisfaction. $H$ is instead determined by a categorical variable $H_i$ and unknown cutoffs $C_1, C_2 \ldots C_n$, and satisfies the $C_1 < C_2 < \ldots < C_{n-1}$ condition. Meanwhile, $X_i$ are the explanatory variables including age, marital status, education, family members, and access to technology, such as TV and the internet; $a$ and $\beta$ are the estimated parameters, and $e_i$ is an error term.

However, unobserved factors may simultaneously affect residents' social capital and subjective well-being indicators (e.g., intrinsic abilities, psychological state, and experience). As a result, in Equation (1), the social capital variable ($S_i$) is potentially endogenous, which we should rule out in the calculation. Previous studies have suggested several approaches to endogeneity, including a two-stage predictor substitution (2SPS) [55,56]. According to Ma, Zheng, Zhu and Qi [55], the 2SPS approach's application is simple and straightforward.

The 2SPS procedures are divided into two stages. First, we examine the social capital model and use the result to create a prediction value of the variables. We adopt the utility maximization theory to determine social capital from the respondents' involvement in organizations. The residents will become members of an association if they think it will benefit them, i.e., making them happier and more satisfied. Hence, we hypothesized that the subjective well-being level is greater if one belongs to more organizations or associations. Since the value of the social capital is censoring (from zero if the resident does not belong to any associations to six if they join all six organizations), we used a Tobit model to estimate the social function, and we specified it as follows:

$$S_i^* = \varphi Z_i + \sigma IV_i + u_i, \text{ and } S_i = min \ 0, \ldots max \ 6) \qquad (2)$$

where $S_i$ represents social capital, captured by the number of organizational memberships; $Z_i$ is the same explanatory variable set in Equation (1); and $IV_i$ is a vector of the instrumental variable. We used neighbors' trust as an instrumental variable, represented by an ordinal value. The respondent's correlation test validates the instrument. It should have a significant correlation on the treatment variable (i.e., social capital) but should not significantly correlate with the outcome variable (i.e., happiness and life satisfaction). Table A1 confirms this regulation. Meanwhile, $\varphi$ and $\sigma$ are the parameters to be estimated, and $u_i$ is an error term. After executing Equation (2), we predicted the social capital variables in the next 2SPS step.

We estimated the impact of social capital on subjective well-being following Equation (2). Before that, the social capital in Equation (1) was replaced with the social capital variable that was predicted in Equation (2). Specifically, we formulated it as follows:

$$H_i^* = \mu Spred_i + \tau X_i + \psi_i, \text{ and } H_i = \begin{cases} 1 \text{ } if \text{ } H_i^* \leq C_1 \\ 1 \text{ } if \text{ } C_1 <; \text{ } H_i^* \leq C_2 \\ n \text{ } if \text{ } C_{n-1} \leq H_i^* \end{cases} \quad (3)$$

where $H_i^*$ and $X_i$ are variables mentioned above, $Spred_i$ is the social capital that was predicted from Equation (2) estimation. $\mu$. and $\tau$ are parameters to be estimated, and $\psi_i$ is an error term for Equation (3).

## 4. Result

### 4.1. Descriptive Statistics

Table 1 presents the descriptive statistics and measurement of the study's variables. The subjective well-being of Indonesian residents shows a relatively good condition since the average value of happiness is 3.036 and life satisfaction 3.323, in line with previous findings by Anna, et al. [57]. The average age is 37.326, with 72.5% being married. A proportion of 36.1% of the total respondents have access to the internet, and 90.8% have television. Only 3.8% of the respondents received no formal education, while 29.5% graduated from primary school and 19.4% from secondary high school. Only 3.6% have an associate degree, and 10.2% have a university degree, i.e., BA, MA, or Ph.D. The average number of family members is 4–5 people, and 51.6% have children under 15 years old. They also reported that their health is relatively good, with an average value of 2.968. The household income is about IDR 1,110,626 per month.

**Table 1.** Descriptive statistics.

| Variables | Measurement | Mean | St. Dev |
|---|---|---|---|
| Social capital | Number of community or association memberships (0–6) | 1.318 | 1.287 |
| Happiness | Self-reported happiness: from 1 = very unhappy to 4 = very happy | 3.036 | 0.498 |
| Life satisfaction | Self-reported life satisfaction: from 1 = not satisfied to 5 =completely satisfied | 3.323 | 0.806 |
| Age | Age of residents in years | 37.326 | 14.932 |
| Marital status | Dummy, 1 for married; 0 otherwise | 0.725 | 0.447 |
| Access to internet | Dummy; 1 if the respondent has access to the internet 0 otherwise | 0.361 | 0.480 |
| No education | Dummy; 1 if the respondent received no education; 0 otherwise | 0.038 | 0.192 |
| Primary education | Dummy; 1 if the respondent's education level is primary school; 0 otherwise | 0.295 | 0.456 |
| Junior education | Dummy; 1 if the respondent's education level is junior high; 0 otherwise | 0.195 | 0.396 |
| Senior education | Dummy; 1 if the respondent's education level is senior high; 0 otherwise | 0.194 | 0.395 |
| Diploma | Dummy; 1 if the respondent's education level is associate degree; 0 otherwise | 0.036 | 0.187 |
| University | Dummy; 1 if the respondent's education level is university level; 0 otherwise | 0.102 | 0.303 |
| Child <15 | Dummy; 1 if the respondent has a child under 15 years old; 0 otherwise | 0.516 | 0.500 |
| Health | Self-reported health: from 1 = very unhealthy to 4 = very healthy | 2.968 | 0.660 |
| Household size | Number of family members in the household (person) | 4.221 | 1.891 |
| Income | Monthly income in Indonesian Rupiah (IDR) per month | 1,110,626 | 984,880 |
| TV ownership | Dummy 1 if the respondent has a TV; 0 otherwise | 0.908 | 0.288 |
| Neighbor trust | Trust in the neighbors: from 1 = strongly disagree to 4 = strongly agree | 2.838 | 0.574 |

Table 2 highlights the differences between urban and rural data. Based on the socioeconomic conditions, residents in urban areas are younger, more educated, have higher income, and better access to technology, such as the internet and TV. Therefore, the effect of social capital in an urban and a rural area might be different. Likewise, based on subjective well-being, urban people are happier and more satisfied than those in rural areas.

**Table 2.** Mean differences of the selected variable base on urban and rural.

| Variables | Urban | Rural | Different |
|---|---|---|---|
| Social capital | 1.249 | 1.419 | −0.170 *** |
| Happiness | 3.054 | 3.011 | 0.043 *** |
| Life satisfaction | 3.353 | 3.280 | 0.072 *** |
| Age | 36.887 | 37.956 | −1.069 *** |
| Marital status | 0.763 | 0.698 | 0.065 *** |
| Access to internet | 0.439 | 0.249 | 0.191 *** |
| No education | 0.023 | 0.061 | −0.038 *** |
| Primary education | 0.231 | 0.388 | −0.157 *** |
| Junior education | 0.177 | 0.220 | −0.043 *** |
| Senior education | 0.215 | 0.164 | 0.051 *** |
| Diploma | 0.048 | 0.020 | 0.027 *** |
| University | 0.132 | 0.058 | 0.074 *** |
| Child < 15 | 0.499 | 0.540 | −0.041 *** |
| Health | 2.966 | 2.971 | −0.005 |
| Household Size | 4.240 | 4.193 | 0.047 ** |
| Income | 1,260,174 | 900,048 | 360,127 *** |
| TV ownership | 0.939 | 0.865 | 0.074 *** |
| Neighbor trust | 2.806 | 2.883 | −0.076 *** |

Note: ** = < 0.05; *** = < 0. 01.

Nevertheless, residents in rural areas have significantly higher social capital (see Figures 1 and 2). As mentioned earlier, there are six community associations in this study. The most popular community association in the urban areas is religious groups with 49.89% membership, followed by voluntary work (21.16%), and community meetings (20.79%). Meanwhile, in a rural area, the most popular membership is religious groups (57.20%), voluntary work (26.28%), and village improvement programs (25.05%). As for cooperatives, only 3.74% of urban people and 2.99% of rural people are members.

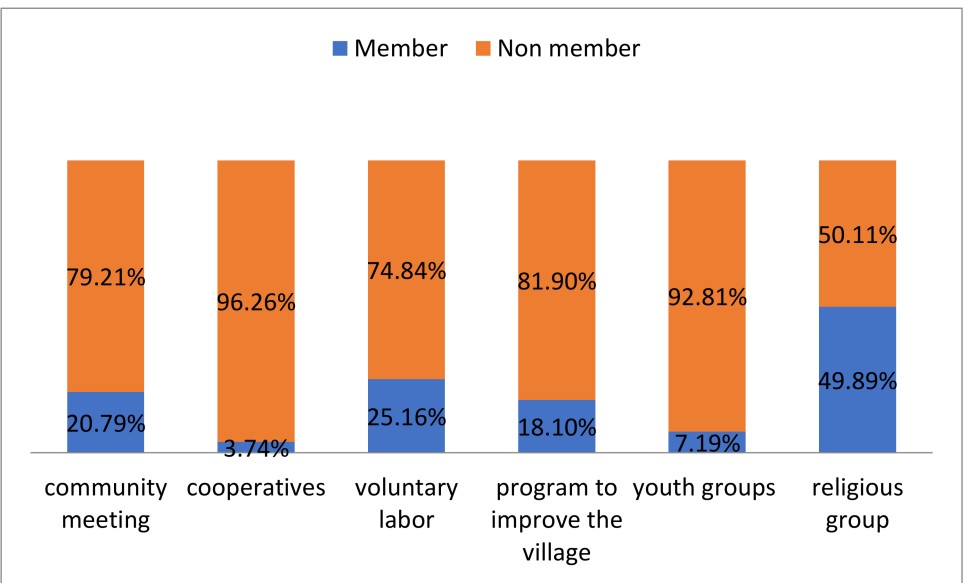

**Figure 1.** Social capital of urban residents.

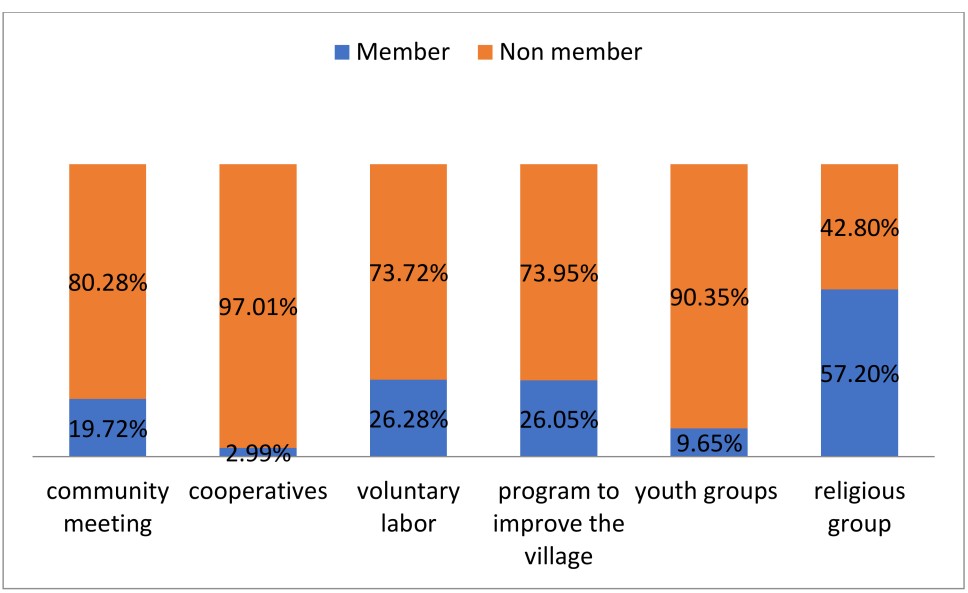

**Figure 2.** Social capital of rural residents.

*4.2. Empirical Results from 2SPS Approach: A Pooled Model*

4.2.1. The Determinants of Social Capital: A Pooled Model

The results of the 2SPS approach used for the full sample are presented in Table 3. The first stage of the 2SPS approach was to estimate the factors that influence residents' social capital, which are specifically diverse, as shown in columns 2 and 3. The results indicate that social capital is positively and significantly influenced by age, marital status, having a child under 15 years old, health status, income, and TV ownership, and it is negatively affected by the education level. For example, age is positive and statistically significant at a 1% level, which means that the older the residents, the more likely they participate in an organization. This finding aligns with a study by Kaase [58], suggesting that people tend to join more organizations as they get older to fill their spare time. Marital status shows a positive and significant effect on social capital. The finding implies that people are more likely to join the community after marriage. Meanwhile, education shows a negative and significant impact on social capital, but the results are counterintuitive. For instance, residents with no education are less likely to join a community association, which may be understandable because they may lack awareness. However, the same is true for those with higher education levels, such as associate degrees. A tentative explanation for why people with an associate degree are not members of an organization is that they use their time for productive activities to earn more income.

**Table 3.** 2SPS—the impact of social capital on subjective well-being: pooled model.

| | First Stage | | Second Stage | | | |
|---|---|---|---|---|---|---|
| **Variable** | **Social Capital** | | **Happiness** | | **Life Satisfaction** | |
| | **Coef.** | **Std. Err** | **Coef.** | **Std. Err** | **Coef.** | **Std. Err** |
| Social capital | | | 0.381 | 0.089 *** | 0.152 | 0.077 ** |
| Age | 0.019 | 0.001 *** | −0.017 | 0.002 *** | −0.007 | 0.002 *** |
| Marital status | 0.096 | 0.022 *** | 0.292 | 0.024 *** | 0.058 | 0.020 *** |
| Access to internet | 0.016 | 0.021 | 0.095 | 0.022 *** | 0.067 | 0.019 *** |
| No education | −0.795 | 0.046 *** | 0.140 | 0.085 | 0.244 | 0.073 *** |
| Primary education | −0.257 | 0.026 *** | −0.023 | 0.035 | 0.017 | 0.030 |
| Junior education | −0.106 | 0.026 *** | −0.009 | 0.029 | 0.062 | 0.024 *** |
| Senior education | −0.051 | 0.026 ** | 0.039 | 0.027 | 0.025 | 0.023 |

**Table 3.** *Cont.*

| Variable | First Stage | | Second Stage | | | |
| | Social Capital | | Happiness | | Life Satisfaction | |
| | Coef. | Std. Err | Coef. | Std. Err | Coef. | Std. Err |
|---|---|---|---|---|---|---|
| Diploma | −0.086 | 0.044 ** | 0.176 | 0.046 *** | 0.072 | 0.039 * |
| University | 0.019 | 0.032 | 0.169 | 0.033 *** | 0.100 | 0.028 *** |
| Child < 15 | 0.093 | 0.019 *** | −0.008 | 0.021 | −0.057 | 0.018 *** |
| Health | 0.049 | 0.011 *** | 0.228 | 0.012 *** | 0.225 | 0.010 *** |
| Household Size | 0.001 | 0.004 | 0.007 | 0.004 *** | 0.011 | 0.004 *** |
| Income | 0.000 | 0.000 *** | 0.000 | 0.000 *** | 0.000 | 0.000 *** |
| TV ownership | 0.135 | 0.026 *** | 0.170 | 0.028 *** | 0.176 | 0.024 *** |
| Neighbor trust | 0.144 | 0.013 *** | | | | |
| _cons | −0.016 | 0.068 | | | | |
| | | | −1.361 | 0.070 | −1.229 | 0.060 |
| | | | −0.347 | 0.067 | −0.137 | 0.058 |
| | | | 2.282 | 0.069 | 1.137 | 0.058 |
| | | | | | 2.722 | 0.060 |
| | | | | | 4.990 | 0.239 |
| Log-likelihood | −48,182.012 | | −19,541.912 | | −19,541.912 | |
| LR chi2(15) | 1655.860 | | 2004.270 | | 2004.270 | |
| Prob > chi2 | 0.000 | | 0.000 | | 0.000 | |
| Pseudo R2 | 0.017 | | 0.049 | | 0.049 | |
| Number of obs | 29,341.0 | | 29,341.0 | | 29,341.0 | |

Note: * = < 0.10, ** = < 0.05, *** = < 0.01.

Having children under 15 years old has a positive and significant coefficient, suggesting that they tend to have higher social capital than those who have not. Health status shows a positive and significant impact on social capital, meaning that those with better health conditions participate in associations more. This finding aligns with a study by Christoforou [52], highlighting the positive association between health conditions and social capital. Income has a positive coefficient and is statistically significant at a 1% level, meaning those with better income are more likely to join associations. Having a stable income allows people to manage their time conveniently, so they may not need a side job or a part-time job. Therefore, they can join community associations more to improve their social capital. This finding supports the study by Maluccio, Haddad and May [16], in South Africa, which highlights a positive association between income and social capital. TV ownership increases the social capital; besides providing entertainment, TV also provides essential information to raise awareness. Lastly, trust in neighbors shows a positive and significant impact on social capital, which means those who trust their neighbors tend to join a community association more. At the same time, the significant result of the neighbor trust variable also confirms that validity is the instrumental variable.

### 4.2.2. The Impact of Social Capital and Control Variables on Subjective Well-Being: A Pooled Model

The second stage of the 2SPS model estimates the impact of social capitals and control variables on subjective well-being (i.e., happiness and life satisfaction). The result is presented in Table 3, specifically columns 4 and 5 for the happiness model, and 6 and 7 for the life satisfaction model. The estimation was obtained using Equation (3) and was executed by an ordered probit model. The result indicates that social capital positively impacts subjective well-being, statistically significant at 1% for happiness and 5% for life satisfaction.

The second stage of the 2SPS approach also seeks the impacts of control variables on subjective well-being indicators, i.e., happiness and living satisfaction. The resident's subjective well-being is positively and significantly affected by marital status, access to

the internet, education, health condition, number of family members, and TV ownership. However, it was negatively and significantly affected by age and having children under 15 years old.

Age has a negative and significant impact on subjective well-being indicators, with the youngest respondents reporting more happiness and life satisfaction than the older ones. This is predictable because they have more energy to accomplish things to be happier and satisfied with daily achievements. This finding supports Nie, Ma and Sousa-Poza [36], highlighting a negative association between age and subjective well-being in China. Marital status positively impacts subjective well-being and is statistically significant at a 1% level for happiness and life satisfaction. It is a common finding in happiness research that marriage improves subjective well-being. For example, Anna, Yusuf, Alisjahbana and Ghina [57] evaluated the determinant of happiness in Indonesia and found a positive relationship between marriage and subjective well-being. According to Stack and Eshleman [59], marriage can promote happiness by improving financial security and health. As for access to the internet, it has a positive coefficient for happiness and life satisfaction too, statistically significant at a 1% level, which is in line with the previous study by Zhu, Ma and Leng [25], Nie, Ma and Sousa-Poza [36], and Zheng and Ma [22].

Most education variables show a positive impact on subjective well-being. However, the most significant education variable on subjective well-being is university education. It has a positive and significant impact on subjective well-being indicators, i.e., happiness and life satisfaction. This finding highlights the importance of higher education, similar to a study by Cuñado and de Gracia [60] in Spain, reporting that people with the highest education tend to be happier and more satisfied. Meanwhile, having children under 15 years old does not significantly impact happiness. However, it has a negative and significant impact on life satisfaction, suggesting that residents who have children under 15 years old report a lower life satisfaction level. A tentative explanation is that they need to juggle raising their children, being productive, and maintaining their budget to take care of their family. In addition, children under 15 years old are in the early stage of puberty, which may give parents extra stress. It shows positive and significant impacts on subjective well-being regarding health conditions, suggesting that better health increases happiness and life satisfaction. Muresan, et al. [61] reported the same findings in European countries. As for income, it has a positive coefficient and statistical significance at 1% for both happiness and life satisfaction. According to Zheng and Ma [22], fulfillment of physical and emotional needs leads to higher well-being, which could easily be achieved if people have enough financial resources. Lastly, TV ownership significantly increases residents' subjective well-being, most likely because TV provides entertainment and information.

*4.3. Empirical Result from 2SPS Approach: An Urban–Rural Model*
4.3.1. The Determinants of Social Capital: An Urban–Rural Model

Like the pooled model, the urban–rural model employs a 2SPS approach, and the result of the first stage is presented in Tables 4 and 5. Overall, the social capital in rural and urban areas is significantly affected by age, education, and TV ownership. However, we present interesting results: marital status, access to the internet, health condition, and income affect the social capital of urban and rural residents differently. We found four interesting points based on the urban–rural typology model. First, the marital status variable has a positive coefficient and is statistically significant at a 1% level for rural areas but not statistically significant for urban areas. This finding suggested that married people in rural areas often have higher social capital than those who are not married. However, marital status does not have that much impact on social capital in urban areas. Second, access to the internet has a significant and positive impact on social capital in rural areas, but it is not the case in urban areas. This suggests that internet access significantly increases residents' social capital only in rural areas. Third, health condition has a positive and significant impact on social capital for urban areas, but it is not the case in rural areas. Fourth, income significantly

increases residents' social capital in urban areas but not in rural areas, probably because urban residents with stable jobs can spare their time to join the community membership.

**Table 4.** 2SPS—the impact of the social network on subjective well-being: urban model.

| Variable | First Stage | | Second Stage | | | |
| | Social Capital | | Happiness | | Life Satisfaction | |
| | Coef. | Std. Err | Coef. | Std. Err | Coef. | Std. Err |
|---|---|---|---|---|---|---|
| Social capital | | | 0.436 | 0.138 *** | 0.314 | 0.119 *** |
| Age | 0.023 | 0.001 *** | −0.019 | 0.003 *** | −0.011 | 0.003 *** |
| Marital status | 0.046 | 0.028 | 0.306 | 0.030 *** | 0.079 | 0.026 *** |
| Access to internet | 0.024 | 0.026 | 0.087 | 0.028 *** | 0.083 | 0.024 *** |
| No education | −0.771 | 0.067 *** | 0.176 | 0.127 | 0.449 | 0.110 *** |
| Primary education | −0.320 | 0.032 *** | 0.023 | 0.056 | 0.092 | 0.048 ** |
| Junior education | −0.148 | 0.032 *** | 0.004 | 0.039 | 0.095 | 0.034 *** |
| Senior education | −0.068 | 0.030 ** | 0.048 | 0.033 | 0.017 | 0.028 |
| Diploma | −0.120 | 0.049 ** | 0.224 | 0.055 *** | 0.093 | 0.046 ** |
| University | −0.017 | 0.036 | 0.202 | 0.038 *** | 0.110 | 0.032 *** |
| Child < 15 | 0.103 | 0.025 *** | −0.006 | 0.029 | −0.076 | 0.025 *** |
| Health | 0.074 | 0.014 *** | 0.207 | 0.018 *** | 0.214 | 0.016 *** |
| Household size | 0.008 | 0.005 | 0.011 | 0.006 *** | 0.014 | 0.005 *** |
| Income | 0.000 | 0.000 *** | 0.000 | 0.000 *** | 0.000 | 0.000 *** |
| TV ownership | 0.128 | 0.039 *** | 0.142 | 0.044 *** | 0.145 | 0.038 *** |
| Neighbor trust | 0.121 | 0.016 *** | | | | |
| _cons | −0.221 | 0.085 *** | | | | |
| | | | −1.436 | 0.082 | −1.223 | 0.070 |
| | | | −0.446 | 0.078 | −0.115 | 0.067 |
| | | | 2.172 | 0.080 | 1.179 | 0.068 |
| | | | | | 2.764 | 0.070 |
| Log-likelihood | −27,586.310 | | −11,469.875 | | −19,912.108 | |
| LR chi2(15) | 1242.150 | | 1093.900 | | 642.880 | |
| Prob > chi2 | 0.000 | | 0.000 | | 0.000 | |
| Pseudo R2 | 0.022 | | 0.046 | | 0.016 | |
| Number of obs | 17,155.0 | | 17,155.0 | | 17,155.0 | |

Note: ** = < 0.05, *** = < 0.01.

**Table 5.** 2SPS—the impact of social network on subjective well-being: rural model.

| Variable | First Stage | | Second Stage | | | |
| | Social Capital | | Happiness | | Life Satisfaction | |
| | Coef. | Std. Err | Coef. | Std. Err | Coef. | Std. Err |
|---|---|---|---|---|---|---|
| Social capital | | | 0.413 | 0.140 *** | −0.001 | 0.120 |
| Age | 0.017 | 0.001 *** | −0.018 | 0.003 *** | −0.007 | 0.002 *** |
| Marital status | 0.141 | 0.036 *** | 0.270 | 0.040 *** | 0.039 | 0.034 |
| Access to internet | 0.099 | 0.037 *** | 0.059 | 0.039 | 0.030 | 0.033 |
| No education | −0.882 | 0.069 *** | 0.236 | 0.142 * | 0.135 | 0.122 |
| Primary education | −0.290 | 0.047 *** | 0.018 | 0.062 | 0.004 | 0.053 |
| Junior education | −0.143 | 0.048 *** | 0.034 | 0.052 | 0.065 | 0.043 |
| Senior education | −0.084 | 0.049 * | 0.061 | 0.051 | 0.065 | 0.042 |
| Diploma | −0.011 | 0.093 | 0.060 | 0.093 | 0.076 | 0.078 |
| University | 0.079 | 0.065 | 0.113 | 0.066 * | 0.116 | 0.055 ** |
| Child <15 | 0.084 | 0.031 *** | −0.024 | 0.032 | −0.043 | 0.027 |

**Table 5.** *Cont.*

| Variable | First Stage | | Second Stage | | | |
| | Social Capital | | Happiness | | Life Satisfaction | |
| | Coef. | Std. Err | Coef. | Std. Err | Coef. | Std. Err |
| --- | --- | --- | --- | --- | --- | --- |
| Health | 0.011 | 0.018 | 0.254 | 0.017 *** | 0.230 | 0.015 *** |
| Household size | −0.008 | 0.007 | −0.001 | 0.007 | 0.001 | 0.006 |
| Income | 0.000 | 0.000 | 0.000 | 0.000 *** | 0.000 | 0.000 *** |
| TV ownership | 0.180 | 0.035 *** | 0.149 | 0.042 *** | 0.181 | 0.036 *** |
| Neighbor trust | 0.146 | 0.021 *** | | | | |
| _cons | 0.233 | 0.113 ** | | | | |
| | | | −1.296 | 0.133 | −1.351 | 0.113 |
| | | | −0.253 | 0.130 | −0.272 | 0.112 |
| | | | 2.398 | 0.132 | 0.977 | 0.112 |
| | | | | | 2.567 | 0.114 |
| | | | | | 4.570 | 0.267 |
| Log-likelihood | −20,422.491 | | −8057.982 | | −14,465.534 | |
| LR chi2(15) | 577.640 | | 887.460 | | 552.570 | |
| Prob > chi2 | 0.000 | | 0.000 | | 0.000 | |
| Pseudo R2 | 0.014 | | 0.052 | | 0.019 | |
| Number of obs | 12,186.0 | | 12,186.0 | | 12,186.0 | |

Note: * = < 0.10, ** = < 0.05, *** = < 0.01.

#### 4.3.2. The Impact of Social Capital and Control Variables on Subjective Well-Being: An Urban–Rural Model

The impact of social capital on subjective well-being based on urban–rural typology is presented in Tables 4 and 5, columns 4 to 7. The finding indicates that social capital positively and significantly affected happiness and life satisfaction in an urban area. However, for the rural model, social capital has a positive and significant impact on happiness only, and it shows an insignificant effect on life satisfaction.

Tables 4 and 5 also present the influence of control variables on subjective well-being based on urban–rural typology. Generally, the finding is similar to the pooled model in Table 3, but the factors affecting the urban and rural residents' subjective well-being are different. For instance, internet access significantly increases the subjective well-being of urban residents, but it does not significantly affect the rural residents. This result may be explained by the fact that internet access has become a primary need in urban areas, so lack of access will deprive them of the information and entertainment (missing out on news, events, experiences, or life decisions that could make one's life better), which they need to feel secure and satisfied. Additionally, the number of family members significantly increases urban residents' subjective well-being, but it does not significantly increase those living in rural areas.

### 5. Discussion

This section discusses this study's main objective: the impact of social capital on subjective well-being based on urban and rural typology mode. This study began by discussing the impact of social capital on subjective well-being from the pooled model, followed by the rural and urban typology models. Additionally, this study makes the first attempt in the estimation strategy by employing a two-stage predictor substitution (2SPS) to address the endogeneity issue that can produce unbiased estimation. This study summarized that social capital has an essential role in improving residents' subjective well-being. With high social capital, people have more relations, trust, and safety in their environment. As such, their happiness and life satisfaction also increase. Residents' subjective well-being is improved by social capital in the form of a community association, notably in terms of happiness and life satisfaction, in terms of buffering community members from mental shocks, such as economic crisis, climate and natural disaster [62,63], health conditions [64],

and food insecurity [65]. The fact that people with the highest social capital are more likely to be happier and more satisfied with life may indicate that they benefit from buffered well-being from such shocks. This is in line with a study in Japan by Hommerich and Tiefenbach [17], whose model estimations show that social affiliation and membership have positive and significant effect on the subjective well-being (i.e., happiness and life satisfaction). Putnam [7] agrees with this, and Winkelmann [66] believes that social capital is a critical determiner of happiness.

Regarding the different effects of social capital in an urban and rural area, this study has revealed that the impact of social capital on subjective well-being is greater for urban areas. There are two tentative explanations. First, the community associations (as the social capital measurement) in urban areas are well-developed, with better infrastructure, technology innovation, and networking. Second, in urban areas, the associations often receive funding from the government in the form of subsidies and training. Helliwell and Barrington-Leigh [67] support this finding, suggesting that social capital would have a different impact in developed regions than in developing regions.

The results of this study have significant theoretical and practical implications. Firstly, the existing literature has found that social capital has a greater impact on rural residents' subjective well-being than that on urban residents [47]. However, past research only covered informal social capital, namely relation with friends, relatives, and neighbors. The empirical findings in our study claim that the subjective well-being impact of social capital—which is measured by community association—is stronger in urban than rural areas. Therefore, the association in rural areas could be improved by mirroring the urban community association. Second, this study offers a valuable insight to promote subjective well-being, not only for Indonesia but also for other countries, since the types of communities explored in this study are not unique to Indonesia: they are common in other developing nations, such as Tanzania [68], China [50,69], India, Madagascar, Cambodia, Kenya, South Africa, and Bangladesh [39]. Third, because social capital strongly promotes subjective well-being, we suggest expanding community associations to give residents access to well-being improvement information and programs, which could bridge the gap between urban and rural areas. Finally, since subjective well-being has been linked to socioeconomic indicators, community association as a social capital indicator could be integrated in programs to reduce inequalities.

One of the study's shortcomings is that it solely used community associations as social capital indicators to quantify the impact of social capital on subjective well-being among urban and rural residents. Other social capital measurements, such as bridging–bonding social capital, could be included in future studies, which can support the findings of the results of this study.

## 6. Conclusions

This study aims to estimate the impact of social capital on subjective well-being based on urban and rural typologies. Two indicators were employed to understand it, i.e., happiness and life satisfaction. This study used large-scale data from 29,341 Indonesian Family Life Survey 5 responses (IFLS-5), consisting of 17,155 urban residents and 12,186 rural residents. The data were analyzed by a two-stage predictor substitution (2SPS) approach to remove the endogeneity barrier that may exist in the estimation. The results indicate that social capital is positively and significantly influenced by age, marital status, having a child under 15 years old, health status, income, and TV ownership, but is negatively affected by education level. In the urban–rural model, we found factors that affect social capital differently. Marital status and internet access significantly increase social capital for rural residents, but do not significantly impact the urban residents' social capital. By contrast, the number of family members and income significantly improve urban residents' social capital, but it is not the case for rural residents.

The findings also reveal that social capital significantly increases happiness and life satisfaction. However, based on the urban–rural model, we find that social capital impacts

subjective well-being differently in urban and rural populations. In the urban model, social capital increases happiness and life satisfaction significantly. However, the rural model indicates that social capital only significantly increases happiness, not life satisfaction.

We believe that developing community associations in rural areas will increase their role in promoting subjective well-being. This can be achieved by investing in infrastructure, training, and technology innovation.

**Author Contributions:** Conceptualizations, methodology, and validation, T.W.N., N.H., H.T. and S.S.; investigation, resources, and data curation, T.W.N. and H.T.; formal analysis and writing—original draft preparation, H.T. and S.S.; writing—review and editing, T.W.N. and N.H.; supervision, N.H. All authors have read and agreed to the published version of the manuscript.

**Funding:** This research received no external funding.

**Institutional Review Board Statement:** Not applicable.

**Informed Consent Statement:** Not applicable.

**Data Availability Statement:** The data presented in this study are available at http://www.rand.org/labor/FLS/IFLS.html (accessed on 4 December 2021).

**Acknowledgments:** The authors are grateful to Brawijaya University.

**Conflicts of Interest:** The authors declare no conflict of interest.

## Appendix A

**Table A1.** Testing the validity of IV.

| IV | Dependent Variables | Correlation Coefficients | *p*-Values |
|---|---|---|---|
| Neighbor trust | Happiness | 0.0071 | 0.2165 |
| | Life satisfaction | 0.0023 | 0.6795 |
| | Social Capital | 0.0777 | 0.000 *** |

Note: *** denotes significance on 1%.

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
