# Peer review of "Promoting Subjective Well-Being among Rural and Urban Residents in Indonesia: Does Social Capital Matter?"

_sustainability, doi:10.3390/su14042375_

Round 1

Reviewer 1 Report

This study aimed to investigate the effects of social capital on subjective well-being based on urban and rural typology, using large-scale data from 29,341 Indonesian residents, comprising 17,155 urban residents and 12,186 rural residents. The topic is interesting. The authors are suggested to make the following modifications to the paper.

Introduction: "We group it into categories:  macro and micro perspectives." what does macro and micro mean?

Research Data: It is suggested to update the data of the paper, so that the results may be more meaningful.

In this study, we divided variables into three categories. how did you get that?

It is suggested that the discussion should be a response to the scientific question of the paper.

Some references to consult:

Farmers’ Satisfaction with Land Expropriation System Reform: A Case Study in China. Land. 2021; 10(12):1353. https://doi.org/10.3390/land10121353 

Contribution of urban ventilation to the thermal environment and urban energy demand: Different climate background perspectives, Science of the Total Environment (2021), https://doi.org/10.1016/j.scitotenv.2021.148791

Author Response

Dear Reviewer:

Thank you for giving me the opportunity to submit a revised draft of my manuscript titled “Promoting subjective well-being among rural and urban residents in Indonesia: Does social capital matter?” to “Sustainability”. We appreciate the time and effort that you and the reviewers have dedicated to providing your valuable feedback on my manuscript. We are grateful to the reviewers for their insightful comments on my paper. We have been able to incorporate changes to reflect most of the suggestions provided by the editorial team. We have highlighted the changes using the track changes function within the manuscript. Pleas see the attachment for our responses.

Kind regards,

Hery Toiba

Agriculture Socio-economic Department, Brawijaya University, Malang, Indonesia

Reviewer 2 Report

The paper has an interesting and timely topic. Here are my comments to improve the paper:

1- in the abstract, make the knowledge gap clearer. I understand you presented a knowledge gap, but it is not very clear what it means. Also, include a discussion of why the rural model indicates that social capital significantly increases happiness, not life satisfaction.

2- similarly, in the introduction, make clearer what knowledge gaps you identified and how your research addresses them. Also, make the research objectives/questions clearer. Answer the “so what?” question. Why investigating such matter is important? End the introduction with an outline of the paper; what comes next?

3- The novelty/originality should be clearly justified that the manuscript contains sufficient contributions to the new body of knowledge from the international perspective.  What new things (new theories, new methods, or new policies) can the paper contribute to the existing international literature? This point must be reasonably justified by a Literature Review, clearly introduced in Introduction Section, and completely discussed in Discussion Section.

4- you need a new section on literature review. You need to acknowledge the existing literature on the issue and clearly identify the knowledge gap. This is an important comment, so seriously think about further engaging with the literature.

5- include some of the recent literature on psychological wellbeing:

https://doi.org/10.3390/su13148086

https://doi.org/10.1016/j.cities.2019.102438 

https://doi.org/10.1371/journal.pone.0174882

6- in a paper that uses a quantitative methodology, you must have two separate sections for ‘results’ and ‘discussions’. The results section simply and objectively reports what you found, without speculating on why you found these results. The discussion interprets the meaning of the results, puts them in context, and explains why they matter.

7- What are the limitations of your methodology/study?

8- you need to refer back to the literature and previous studies in your result, discussion and conclusion sections.

Author Response

(The authors gave the same response as above.)

Round 2

Reviewer 1 Report

The paper has been improved, but there are still problems. 

The introduction should further highlight the scientific problems, motivations, and possible innovations of the paper. 

Literature Review is still not sufficient.

Discussion:The contribution of this study is still unclear, and the author is still emphasizing what they have done rather than stating the value of it. 

Author Response

Dear Reviewer,

Thank you for giving us the opportunity to submit a revised draft of our manuscript. We appreciate the time and effort that you and the reviewers have dedicated to providing your valuable feedback on my manuscript. We are grateful to the reviewers for their insightful comments on our paper. We have been able to incorporate changes to reflect most of the suggestions provided by the reviewers. We marked up using the “Track Changes” function within the manuscript.

Best regard,

Hery Toiba

Reviewer 2 Report

Thank you for addressing my comments. Your literature review section is too short and you will need to further discuss the relevant literature to your research.

Author Response

Dear Reviewer,

Thank you for giving me the opportunity to submit a revised draft of my manuscript. We appreciate the time and effort that you and the reviewers have dedicated to providing your valuable feedback on my manuscript. We are grateful to the reviewers for their insightful comments on my paper. We have been able to incorporate changes to reflect most of the suggestions provided by the reviewers. We marked up using the “Track Changes” function within the manuscript.

Best regard 

Hery Toiba

Round 3

Reviewer 1 Report

The revised version is fine.